# ESBL Displace: A Protocol for an Observational Study to Identify Displacing *Escherichia coli* Strain Candidates from ESBL-Colonized Travel Returners Using Phenotypic, Genomic Sequencing and Metagenome Analysis

Michael Schweitzer [1], Alfredo Mari [1,2,3], Tim Roloff [1,2,4,5], Esther Künzli [6], Stefanie Heller [1], Diana Albertos Torres [1,4], Marco Meola [1,2,4,5], Danica Nogarth [1], Leanid Laganenka [7], Lisa Prampolini [6], Helena M. B. Seth-Smith [1,2,4,5], Olivia Grüninger [1], Alexander Gensch [1], Josiane Reist [1], Sebastian Bonhoeffer [8], Wolf-Dietrich Hardt [7,†] and Adrian Egli [1,4,7,*,†]

1   Applied Microbiology Research, Department of Biomedicine, University of Basel, 4001 Basel, Switzerland
2   Swiss Institute of Bioinformatics, 4031 Basel, Switzerland
3   Personalized Health Basel, University of Basel, 4001 Basel, Switzerland
4   Institute of Medical Microbiology, University of Zurich, 8006 Zurich, Switzerland
5   Clinical Bacteriology and Mycology, University Hospital Basel, 4031 Basel, Switzerland
6   Swiss Tropical and Public Health Institute, 4051 Basel, Switzerland
7   Institute of Microbiology, Department of Biology, ETH Zurich, 8092 Zurich, Switzerland
8   Department of Environmental Systems Science, ETH Zurich, 8092 Zurich, Switzerland
*   Correspondence: aegli@imm.uzh.ch; Tel.: +41-44-634-26-60
†   These authors contributed equally to this work.

**Abstract:** Introduction: Invading extended-spectrum beta-lactamase-producing *Escherichia coli* (ESBL-PE), non-ESBL *E. coli*, and other bacteria form a complex environment in the gut. The duration and dynamics of ESBL-PE colonization varies among individuals. Understanding the factors associated with colonization may lead to decolonization strategies. In this study, we aim to identify (i) single *E. coli* strains and (ii) microbiome networks that correlate with retention or decline of colonization, and (iii) pan-sensitive *E. coli* strains that potentially could be used to displace ESBL-PE during colonization. Methods and analysis: We recruit healthy travellers to Southeast Asia for a one-year prospective observational follow-up study. We collect and biobank stool, serum, and peripheral blood mononuclear cells (PBMCs) at predefined timepoints. Additional information is collected with questionnaires. We determine the colonization status with ESBL-PE and non-ESBL *E. coli* and quantify cell densities in stools and ratios over time. We characterize multiple single bacterial isolates per patient and timepoint using whole genome sequencing (WGS) and 16S/ITS amplicon-based and shotgun metagenomics. We determine phylogenetic relationships between isolates, antimicrobial resistance (AMR; phenotypic and genotypic), and virulence genes. We describe the bacterial and fungal stool microbiome alpha and beta diversity on 16S/ITS metagenomic data. We describe patterns in microbiome dynamics to identify features associated with protection or risk of ESBL-PE colonization. Ethics and dissemination: The study is registered (clinicaltrials.gov; NCT04764500 on 09/02/2019) and approved by the Ethics Committee (EKNZ project ID 2019-00044). We will present anonymized results at conferences and in scientific journals. Bacterial sequencing data will be shared via publicly accessible databases according to FAIR principles.

**Keywords:** extended spectrum beta-lactamase; *Escherichia coli*; sensitive; sequencing; screening; travel; displacing; colonization; competition; metagenomics

## 1. Introduction

Infections with extended spectrum beta-lactamase-producing *E. coli* (ESBL-PE) are associated with high morbidity [1–3], mortality [2,3], and healthcare costs [2,3]. It is crucial

to reduce its ongoing spread between humans, animals, and the environment. A critical element to consider is the identification of the potential source to break the chain of transmission. In hospitals, patients colonized with multidrug-resistant bacteria are often isolated with special hygiene measures, which is costly and is associated with longer hospital stays and a higher mortality risk [4,5]. Although special contact precautions do not seem to play a major role in limiting the spread of ESBL-PE in healthcare settings, they may be important in high-risk situations with carriers of more virulent strains such as *E. coli* ST131 [6,7]. Attempts of decolonization with antibiotic treatment are mostly unsuccessful and only work for individual bacterial species [8]. In this study, we focus on ESBL-producing Enterobacteriaceae because they are among the priority pathogens according to the list of antibiotic-resistant bacteria for which new antibiotics are urgently needed. This list was published by WHO in 2017 [9]. The main contributor to the global spread of ESBL-PE is *E. coli*, as it is the main vector of this resistance mechanism in a community setting [10–12]. Therefore, new approaches to decolonize patients and reduce the burden of antimicrobial resistance would have a strong impact on clinical and epidemiological management.

Travellers returning from endemic areas are at a high risk of being colonized or infected with ESBL-PE [13–17]. Travel returner cohorts allow us to study factors linked to colonization, duration of colonization, and the resulting impact on the gut microbiome. High prevalence for colonization with ESBL-PE was documented in travellers returning from India, reaching up to 80% [15,16,18,19], or South and Eastern Asia, Northern Africa, and Central and South America [13,17,18,20,21]. The post-travel colonization status has been linked to person-to-person contact [22–24], taking antibiotics, gastrointestinal symptoms during travel [19], age [18], and consumption of food from street vendors [17]. Furthermore, the risk of being colonized with ESBL-PE potentially depends on the composition of the gut microbiota. Dysbiosis, due to antibiotics, could be associated with a reduction in colonization resistance, which is a well-established concept in the case of *C. difficile* [25] or *Salmonella enterica* serovar Typhimurium infection [26]. The duration of colonization is variable [15–17] and little is known about microbiological factors related to the dynamic changes.

In this study, we investigate the dynamic changes in ESBL-PE colonization in healthy travel returners from Southeast Asia. We collect strains and stool samples for this observational study prospectively to comprehensively characterize individual ESBL-positive and -negative *E. coli* strains. We characterize the samples by phenotyping and whole genome sequencing (WGS) and determine the diversity of the gut microbiome using 16S/ITS metagenomic sequencing. We hypothesize that certain pan-sensitive *E. coli* strains may be ingested randomly via food consumption or from other environmental reservoirs. Once a competitive strain is ingested, it will outcompete and displace ESBL-PE in colonized patients. These strains are assessed using a mouse model to determine their displacement potential. The results of mouse-related strains will not be discussed in this study protocol, which focuses on the clinical study in humans. This could lead to new therapeutic applications of displacing *E. coli* strains in long-term colonized individuals and may help to reduce the spread of multidrug-resistant bacteria.

## 2. Aim and Objectives

Our main objective is to investigate the interplay of ESBL-PE and pan-sensitive *E. coli* within the context of the gut microbiota. We aim to better understand the factors influencing the colonization status with ESBL-PE.

*The Detailed Aims Include*

The recruitment of travellers (*n* = 40) for a prospective observational study with 11 timepoints over a 52-week follow-up. The generation of a biobank with single bacterial strains including ESBL-PE and sensitive *E. coli*, stool samples, serum, and peripheral blood mononuclear cells (PBMCs).

To better understand the transmission risks between humans, animals, and the environment, we use data from our detailed questionnaires, each at the time of sampling

(see Supplementary Files S1–S3). However, this is not the main focus of our study. Our goal is to better understand which factors within an individual favour or hinder ESBL-PE colonization by evaluating the following two characteristics:

Phenotypic characterization: the description of absolute and relative changes between ESBL-PE and sensitive *E. coli* strains per participant over time. Identification and description of participants without, with short, or with long colonization in the context of demographic, clinical, microbiological, and immunological factors.

Genotypic characterization: the analysis of ESBL-PE and sensitive *E. coli* genomes using whole genome sequencing including phylogenetic and plasmid analysis, and the determination of antimicrobial resistance and virulence genes. Description of within-host evolution of single strains and dynamic changes within the gut microbiome using 16S/ITS metagenomics for alpha/beta diversity, linkage to dysbiosis events, and community assessment through microbial networks analysis over time.

## 3. Patients and Public Involvement

The study proposal was evaluated by the local ethical committee based on the Human Research Act of Switzerland (EKNZ project ID 2019-00044 on 12 February 2019, see Supplementary File S4). The study is registered (clinicaltrials.gov; NCT04764500 on 09/02/2019). This is an observational prospective study without intervention in humans. All study participants in this prospective study received detailed information and provided written informed consent.

## 4. Methods/Design

Setting. The study is conducted in healthy travellers going to Southeast Asia. The recruitment period starts prior to the COVID-19 pandemic, and all participants had the first samples collected between June and December 2019.

Study design. This study is designed as a monocentric observational prospective cohort study. Patient recruitment is coordinated via the Swiss Tropical and Public Health Institute (Basel, Switzerland). We collect samples one week before the travel episode and at a 52-week follow-up with additional sampling at 10 predefined time points, resulting in a total of 11 samples per participant. Sample processing is performed at the Applied Microbiology Research Laboratory of the University of Basel.

Recruiting. Around 14,000 people per year consult the Swiss Tropical and Public Health Institute in Basel for pre-travel health advice. We ask travellers going to Southeast Asia to participate during pre-travel counselling. Inclusion criteria: (i) Travel to Thailand, Vietnam, Nepal, Cambodia, India, Myanmar, Laos, Bhutan, and Bangladesh; (ii) age of 18 years or older; and (iii) a maximum four-week travel duration. Exclusion criteria: (i) Diagnosis of inflammatory bowel disease [27]; (ii) immunosuppression, e.g., due to rheumatic disease or organ transplantation; and (iii) antibiotic use at the first sampling time point.

Patient and Public Involvement. Patients are involved before travelling to one of the included travel destinations. The patients are contacted and informed in detail about the study. As this was an observational study, we inform patients about the results of ESBL colonization if desired, according to the ethical protocol. We plan to disseminate the results in a scientific conference and thereby inform the public. However, it is not appropriate or possible to involve patients or the public in the design, conduct, reporting, or dissemination plans of our research.

Sampling and data collection. After study inclusion, each participant receives a portable camping freezer to store stool samples at $-20\,^\circ$C immediately after collection. We ask participants to provide frozen stool samples and questionnaires as soon as possible after collection. Figure 1 provides an overview on the study design and sampling strategy. Stool samples are collected at week $-1$ (one week before travel), week 0 (on travel return), weeks 2, 4, 6, 8, 10, 12, 16, 20, and 52. Based on a previous published study [17], we knew that the ESBL-PE colonization status shows a strong dynamic within the first weeks after

travel return. Therefore, we hypothesized that the greatest changes in the microbiome and colonization status would likely occur around the time of travel. We assumed that the microbiome would gradually stabilize after the trip and that relevant changes would then become less likely. We collect a total of 55 mL of blood at week $-1$, 0, 6, 12, and 20 in Serum and CPT tubes (Becton Dickinson, Allschwil, Switzerland) for later analysis of humoral- and cellular *E. coli*-specific immunity. Participants also consent to determining the single nucleotide polymorphisms associated with host immunity.

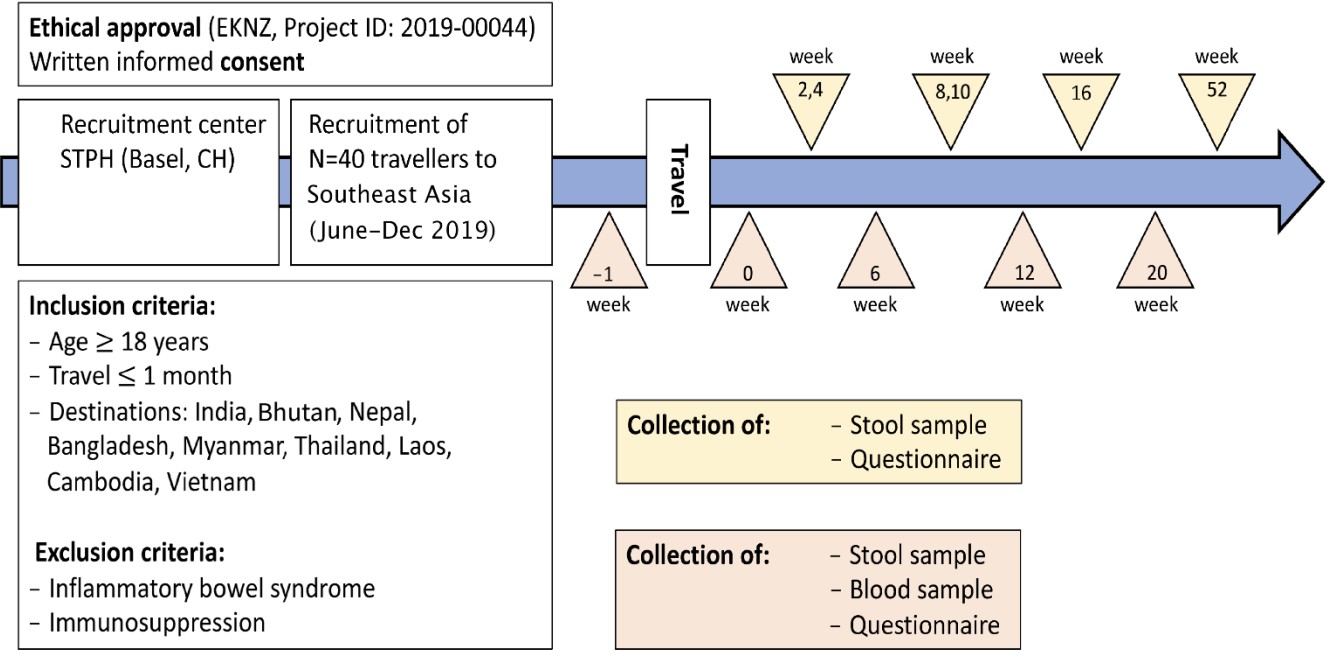

**Figure 1.** Study flow for patient recruitment and sample collection.

Blood sample processing. We aliquot serum (5 mL) for antibody measurement and freeze it at $-75\,^{\circ}\mathrm{C}$ until further usage. We process CPT tubes (50 mL) for PBMC isolation according to the manufacturer's instruction. The PBMC remnant is then dissolved and frozen in a freezing medium (90% heat-inactivated fetal bovine serum and 10% Dimethyl-sulfoxid) with a cell concentration of $5 \times 10^6$ per ml of freezing medium and stored in liquid nitrogen ($\mathrm{N_2}$). The analysis of PBMCs will be performed at a later timepoint and no specific protocol has been developed at this stage.

Stool sample processing. The stool collection tubes (Eppendorf, 2 mL) contain Tryptic Soy Broth (TSB, Merck & Cie, Schaffhausen, Switzerland) and 15% glycerol. The stool sample is added in a 1:1 ratio. The collected stool samples are immediately stored at $-20\,^{\circ}\mathrm{C}$. Stored stool samples are processed as follows: thawed samples are slightly vortexed, then samples are aliquoted and stored at $-75\,^{\circ}\mathrm{C}$. We inoculate a 10 µL loop from the original sample onto two selective agar plates: a MacConkey II agar plate (Becton Dickinson) and a ChromID-ESBL agar plate (bioMérieux SA, Petit-Lancy, Switzerland) to isolate single bacterial colonies. In cases without growth of *E. coli*, we perform an enrichment of the sample in BBL Enriched Thioglycoate Medium (Becton Dickinson) before re-plating onto selective plates. The media are incubated under aerobic conditions for 18–24 h at $36\,^{\circ}\mathrm{C}$.

### 5. Phenotypic Characterisation

Resistance mechanisms. On the second day, 15 colony forming units (CFU) from each MacConkey II plate are sub-cultured on a Columbia agar +5% sheep blood plate (Becton Dickinson), and in parallel on a ChromID-ESBL plate as a control to obtain only ESBL-negative *E. coli*. From the ChromID-ESBL plate, eight single colonies are sub-cultured onto a blood agar plate to obtain ESBL-producing *E. coli*. On the third day, the bacterial

species is confirmed by Matrix-Assisted Laser Desorption/Ionisation Time-of-Flight Mass Spectrometry (MALDI-TOF MS) [28]. Figure 2 provides an overview of the analytical workflow. ESBL production is confirmed for one representative isolate per time point with the MASTDISC diffusion test (Mast Diagnostica, Reinfeld, Germany) on a Mueller-Hinton II agar (Becton Dickinson). We perform and read the test according to the European Committee on Antimicrobial Susceptibility Testing (EUCAST) guidelines [29]. Whole genome sequencing (WGS) of each isolate allows for molecular confirmation and identification of the ESBL genes.

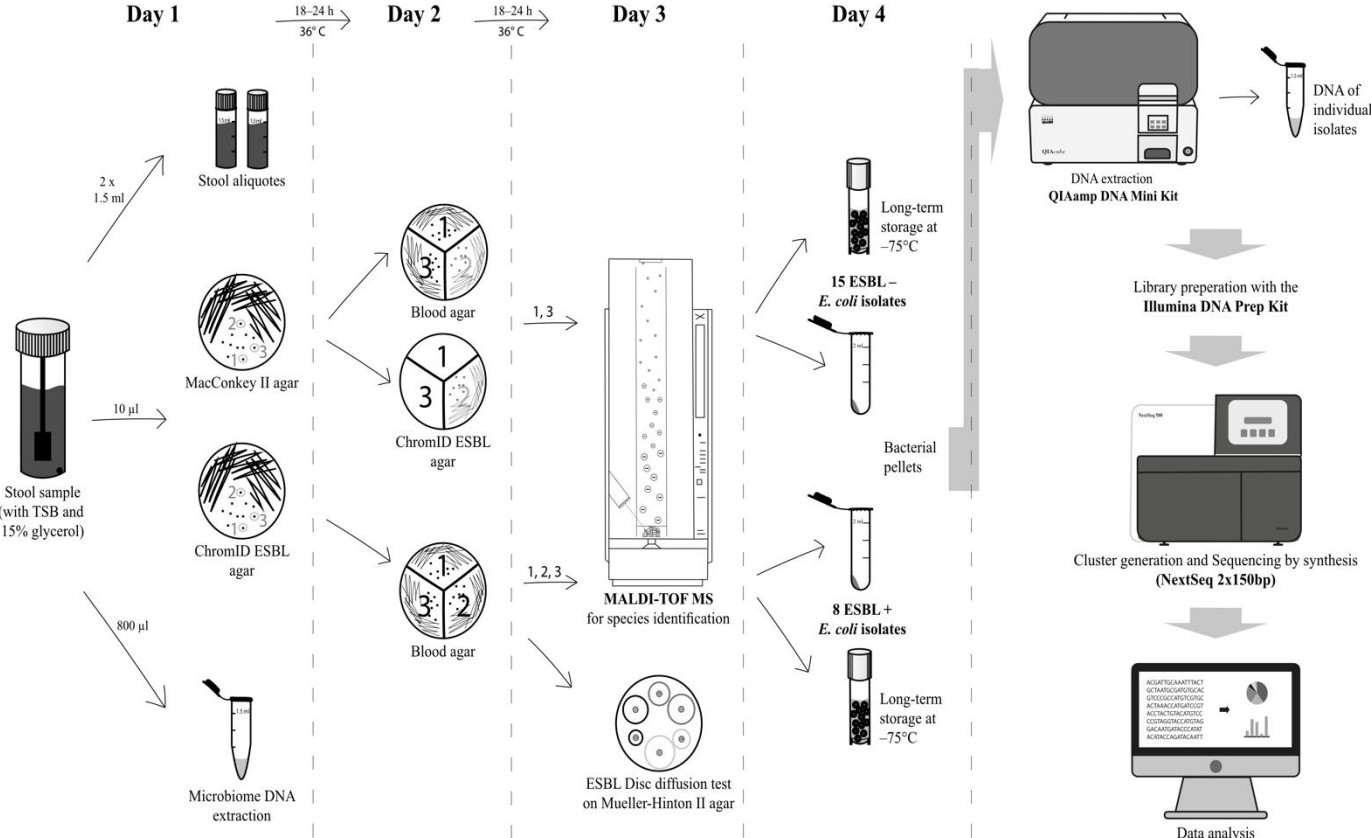

**Figure 2.** Microbiological processing of samples.

Storage of isolates. We transfer the confirmed ESBL-PE and ESBL-negative *E. coli* isolates into a cryotube containing a specially formulated cryopreservation (composition details are not provided by the company) solution with beads for long-term storage at −75 °C (Microbank Pro-Lab Diagnostics, Richmond Hill, Canada).

## 6. Genotypic Characterisation

DNA extraction from single isolates. We extract the DNA from sub-cultured individual *E. coli* isolates for subsequent genomic characterisation. We use the QIAamp DNA Mini Kit (Qiagen, Hombrechtikon, Switzerland) on a Qiacube robotic system (Qiagen, Hilden, Germania) according to the manufacturer's instructions. We measure DNA concentration by fluorometric quantification with the Qubit system (Thermo Fisher Scientific, Reinach, Switzerland). DNA is stored at −75 °C until further processing.

DNA extraction from stool samples. We extract the DNA from the whole stool sample using the QIAamp Power Fecal Pro DNA Kit (Qiagen). Briefly, 800 μL samples are homogenized and processed according to the manufacturer's instruction, including the bead-containing tube (PowerBead Pro Tube from Qiagen, Hilden, Germania). After bead beating, we centrifuge the tube at 10,000× *g* for 30 s, remove the supernatant, and add 800 μL of CD1 lysis buffer solution. The bacterial cells are then mechanically and chemically

lysed for 10 min at 25 Hz in a TissueLyser (Qiagen, Hilden, Germania). We spin the beads at 15,000× *g* for 1 min and then use 600 µL of the supernatant to extract the DNA (Qiagen, Hilden, Germania). We use obtained DNA for 16S/ITS metagenomic sequencing.

Real-time PCR for specific resistance genes. In cases with phenotypic evidence or suspicion of carbapenemase production (positive ESBL screening on ChromAgar plate, but no evidence for ESBL in MAST-DISC assay), the easyplex SuperBug CRE test (AmplexDiagnostics, Gars am Inn, Germany) is performed. The assay is used to detect KPC, NDM, OXA-48, OXA-181, and VIM carbapenemases, and the ESBL genes CTX-M-1 and CTX-M-9 groups [30].

PCR to quantify ESBL genes. We use whole stool DNA to quantify the amount of ESBL genes relative to *E. coli* bacteria for each sample. Various protocols will be evaluated.

Whole genome sequencing (WGS). We characterize and compare collected *E. coli* isolates using WGS. We describe the phylogeny and identify possible resistance and virulence genes as previously described [31]. For the library preparation, we use the Illumina DNA Prep Kit (Illumina, Zürich, Switzerland) with 100 ng total DNA input on a Hamilton Star robotic system (Hamilton, Germany). We sequence libraries using a NextSeq 500 platform (Illumina), providing 150 nt paired-end reads. Sequences are quality controlled according to our ISO-accredited (ISO/IEC 17025) pipeline and pass QC standards with a minimum average coverage of 40-fold.

16S/ITS sequencing. Prior to 16S library preparation, the extracted DNA is diluted to 1 ng/µL. Next, we prepare libraries using the QIAseq 16S/ITS Screening Panel kit (Qiagen, Hilden, Germania) and according to the manufacturer's instructions, a two-stage PCR workflow. First, for each DNA sample, three separate PCR reactions are prepared to generate six different amplicons that cover the entire 16S gene. Each PCR reaction contains region-specific primer pairs: panel pool 1 (V1/V2 and V4/V5), pool 2 (V2/V3 and V5/V7), and pool 3 (V3/V4 and V7/V9) (QiaSeq, Qiagen, Hilden, Germania). We use the QIAseq 16S/ITS PCR reaction setup and adapt it for 20 cycles due to low bacterial content. Following the first PCR reaction, we add 20 µL UCP PCR water (UCP Qiagen kit, Hilden, Germania) to each of the three PCR reactions, which are then pooled. Following PCR purification with QIAseq beads, the indices required for sequencing are added to the purified PCR product. Next, we performed a clean-up after the PCR reaction using QIAseq beads. We measure the final product DNA concentration using Qubit. The molarity of each library is determined as indicated below in Figure 3 considering an average library size of 550 bp:

$$\frac{(\text{concentration in ng/µl})}{(660 \text{ g/mol} \times \text{average library size in bp})} \times 10^6 = \text{concentration in nM}$$

**Figure 3.** Calculation to determine the molarity of each library.

We dilute samples to 2 nM, pool them, and use them as input for the denaturation procedure prior to sequencing. For denaturation, we add 5 µL of freshly prepared NaOH (0.2 N) to 5 µL of the pooled library and incubate for 5 min at room temperature. Next, we add 5 µL of Tris 200 mM for NaOH neutralisation. A library concentration of 17 pM is used for sequencing. The MiSeq Reagent Kit v3 (Illumina) 600 cycle is used for paired-end 276 bp sequencing.

## 7. Questionnaires

Questionnaires are sent to each participant at each time point of the stool sample collection (see Supplementary Files S1–S3). The questionnaire has been adapted from a previous study [15]. The questionnaires cover travel-related metadata, e.g., travel-related medical problems, exposure to the healthcare system, antibiotic consumption, behaviour during travelling, etc.

## 8. Data Analysis

We use descriptive statistics to explore the study population and the distributions of ESBL-PE rates. We use the median and interquartile range unless otherwise stated. Non-parametric tests such as the Mann–Whitney U test are used for comparisons between groups. Chi-squared tests are used for categorical comparisons. Statistics contain corrected data for multiple comparisons.

Bioinformatics of whole genome sequencing. We process sequencing reads through our in-house bioinformatic pipeline, which consists of an initial quality checking and filtering. This includes the trimming of sequencing adaptors. All reads with a Phred score of less than 12 are truncated using a sliding window approach over 4 base pairs. Next, all reads shorter than 100 bp are discarded. Initial quality checking and filtering of the reads not meeting the length threshold is performed using Trimmomatic [32].

Reads are assembled through the unicycler pipeline (version v0.3.0b) using default options [33]. We process the bioinformatics as described in the next paragraphs, however, we will adapt this to the current version of required tools. The assemblies are thereafter annotated using Prokka [34] and their quality is assessed by the Busco pipeline [35]. Additional mapping is performed to double-check the correctness of the assembly and to evaluate the coverage information. Reads are mapped against the assembly using the Burrows–Wheeler algorithm [36] and the assembly is polished using Pilon [37].

For phylogenetic analysis, we use SeqSphere+ (Ridom, Münster, Germany) to determine the core genome multi-locus sequencing type (cgMLST) and visualize trees including all ESBL-PE and non-ESBL *E. coli* for each participant.

For the prediction of resistance and virulence genes, we annotate the genes with Abricate (https://github.com/tseemann/abricate, version 0.8.7, accessed on 19 August 2018) and AMRfinderplus [38]. For the former, the NCBI, Resfinder, CARD, and VFDB databases are used [39–42]. Later pipeline improvements will include hAMRonisation (https://github.com/pha4ge/hAMRonization, not yet accessed). We also use a whole-genome approach to describe the dynamics within each host, focusing on genomic and plasmid changes. Plasmid typing and detection is conducted directly on the assembly using MOBsuite [43]. As the software provides a separation of the assembly in plasmid and chromosomal contigs, an additional AMR gene search is performed on the grouped contigs in the same way as described above, thereby allowing us to determine if a resistance gene is on a plasmid or chromosome. We use distance-matrices approaches to compare group strains within a single patient.

Bioinformatics of metagenome analysis. The 16S/ITS reads are processed according to the DADA2 pipeline [44] embedded in the QIIME2 plugin [45], using the pseudo-pooling strategy to provide additional robustness to the ASV (amplicon sequence variant) call. Full-length ASV clusters are obtained by matching the individual ASVs to a reference-based phylogeny using the fragment insertion technique, still within the QIIME2 platform. To ensure sample comparability, the ASV tables are resampled through rarefaction at the saturation point of alpha diversity; alternatively, this step can be removed and replaced by a simple normalization step instead. Alternatively, if the saturation point results in the loss of multiple samples, manual thresholding is performed. Diversity analysis is conducted using the chao1 metric for alpha diversity [46] and binary/weighted Bray–Curtis dissimilarity for beta diversity [47]. Finally, a principal component analysis for each patient is performed. Rarefaction, diversity analysis, and principal component analysis are conducted within the QIIME2 platform and R [48].

Network analysis is performed as ASV/cluster co-occurrence through multivariate regression using both the SCNIC plugin [49] and the CoNET platform [50]. The former approach ensures an adequate treatment of sparse and compositional data, with a focus on finding highly connected modules, and the latter ensures a consensus approach of several dissimilarity metrics and distances, resulting in a bootstrapped and therefore robust output in terms of co-occurrences. When using CoNET, in order to ensure unbias consensus correlation metrics, and account for complementary handling of outliers and

sample size, the Pearson correlation coefficient, Spearman rank correlation coefficient, Kendall distance, Kullback–Leibler distance, and Bray–Curtis dissimilarity are used. The $p$-values and adjusted $p$-values are considered significant if they are less than 0.05.

## 9. Preliminary Results

Recruitment. We enrolled a total of 40 participants. Two participants dropped out before the initial sample collection. One due to the cancellation of the trip and one without providing any reason. We noted that a slight majority are female study participants (21/38, 55%). The median age is 40 years (range: 22–77 years). The countries visited ($n$ = visits per country) included: Thailand ($n$ = 13), Vietnam ($n$ = 9), Nepal ($n$ = 7), Cambodia ($n$ = 6), India ($n$ = 5), Myanmar ($n$ = 3), and Laos ($n$ = 2). Some participants visited more than one country and two travellers did not provide details on national destinations.

Sample processing. To date, we have completed the processing of stool samples from 35 participants. From these, we have isolated a total of 3937 ESBL-negative *E. coli* and 688 ESBL-PE. Determination of the ESBL genes present is pending.

Three travellers stopped delivering material at week 8, week 10, and week 52 respectively. One participant due to the onset of the COVID-19 pandemic and two for unknown reasons (loss during follow-up).

Data on colonization status after travelling. Only 1/38 (3%) was colonized with ESBL-PE prior to travel. Our interim analysis showed that 63% (24/38) of all participants were colonized with ESBL-PE on the first study visit after the travel (week 0). For 3 participants, data collection was incomplete (dropouts) and some samples were missing at the time of writing this report (missing data). However, since we expected that the microbiota would change the most around the time of travel, we considered it advisable not to exclude the samples of these three individuals from further investigation (Figure 4). The colonization rates according to destinations were as follows: India (5/5), Vietnam (8/9), Cambodia (5/6), Nepal (5/7), Laos (1/2), Thailand (5/13), and Myanmar (0/3).

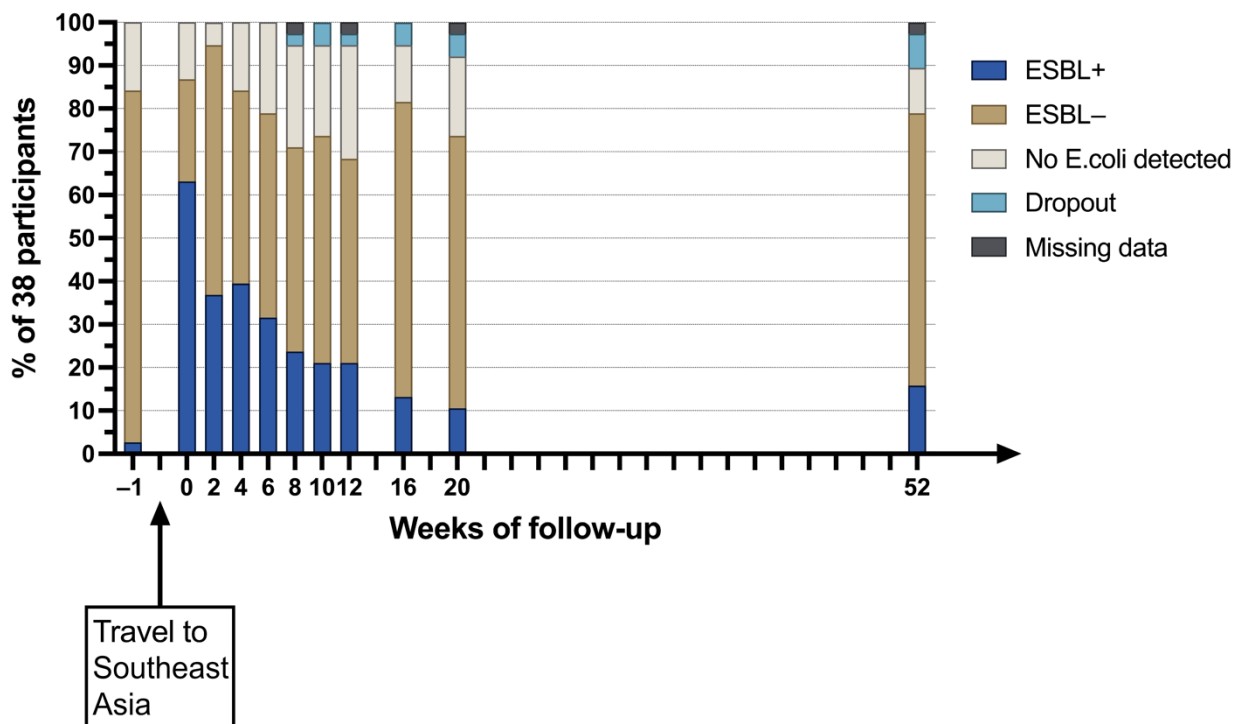

**Figure 4.** Preliminary results on colonization status during follow-up.

## 10. Discussion

Our study focuses on (i) describing colonization dynamics in healthy individuals newly colonized with ESBL-PE and (ii) identifying factors associated with no, short-, or long-term colonization. We use a healthy cohort of travel returners from highly endemic countries in Southeast Asia. In general, international travel is a main risk factor for acquiring ESBL-PE. However, a systematic review found that travellers in Asia more often showed long-term colonization, while multidrug-resistant *E. coli* (MDR *E.coli)* was associated with a lower risk compared to other MDR species [13]. A study from the Netherlands found an association of sustained ESBL-PE carriage and CTX-M-9 group ESBL genes, and reported longer colonization rates by ESBL-producing *E. coli* compared to ESBL-producing *K. pneumoniae* [17].

Our study includes 38 healthy individuals who were followed for one year, starting one week before and ending 52 weeks after the travel episode. We found that only a minority of 3% were colonized prior to travelling. This is comparable to results of two other studies, one conducted in Sweden and the other in Switzerland, in which 2.4% and 2.8% of travellers tested positive before travelling, respectively [15,18]. However, a study from the Netherlands from 2013 found that 8.6% were colonized before travel [20]. Preliminary data from our cohort shows a 21% rate of long-term colonization. In a systematic review and meta-analysis from 2016, the colonization rates after 12 months of MDR-*E. coli* carriage in community residents was 25.4% [51].

In our study, we will also explore the dynamic of the gut microbiome over time. The composition of the gut microbiome may be critical in understanding the role of "invasive" species such as ESBL-PE into a non-ESBL *E. coli* habitat. Previous studies have shown that the integrity of the gut microbiome is fundamental to colonization resistance [25,52,53]. It has also been shown that the microbiome can be altered by factors such as the environment, diet, age, hospitalization, and antibiotic use [54]. This variability is the reason for the possible acquisition of multidrug-resistant bacteria, but it may also be used as a novel strategy to increase colonization resistance and decolonize patients. Probiotics could be an easy-to-use and a cost-effective way to achieve this goal [55,56]. In 2011, Tannock et al. investigated the decolonization ability of the Nissle 1917 *E. coli* strain in elderly residents of long-term care facilities who were colonized with MDR *E. coli.* The application of this strain showed no success in decolonization [57]. However, it is very likely that different species and strains may have the ability to displace MDR bacteria successfully. Promising results in this regard were explored in a review that examined the use of probiotics for the prevention of antibiotic-associated diarrhoea in children [58]. A way to improve the probiotic effect may also be to use a combination of several probiotic strains or a symbiotic product containing a combination of probiotics and prebiotics [59]. A study from Stecher et al. has shown that bacterial species are more likely to colonize when closely related species already occupy a niche [60]. This finding underlies our experimental plan to find pan-sensitive *E. coli* strains that can displace ESBL-PEs due to fitness advantages. In the absence of selection pressure from antibiotics, ESBL-PEs are likely to be at a disadvantage due to the resistance-fitness trade-off [61].

## 11. Strengths and Limitations of the Study

The strength of this study lies in the relatively large number of participants (*n* = 40) who are followed over a one-year period with multiple sampling time points to capture the rapid dynamics of intestinal colonization in healthy travellers.

The second strength is the large collection of single bacterial strains and microbiome samples, which provides the necessary granularity to study the dynamics of ESBL-PE interactions with the gut microbiome and specifically with pan-sensitive *E. coli*.

The third advantage consists of the collection of serum and PBMCs, which allows us to assess the humoral repertoire in the context of microbiological changes.

A challenge of the study is that not every participant will be colonized with ESBL-PE. As a main aim was to study the reasons for colonization and de-colonization over time,

a higher initial colonization rate would provide us more samples to study this. It will, however, also generate critical knowledge on the microbiota network providing potential resilience towards colonization with ESBL-PE.

A second limitation is that many factors potentially influence the gut microbiome and its stability such as diet, antibiotic exposure, host genetics, etc. The complexity of these changes can only be partially captured with the available samples.

A third limitation is that the clinical study can only identify candidates potentially capable of decolonization (i.e., pan-sensitive *E. coli*) by time-wise association. However, this will subsequently be further evaluated and confirmed in the mouse model experiment.

**Supplementary Materials:** The following supporting information can be downloaded at: https://www.mdpi.com/article/10.3390/microbiolres14010015/s1, Files S1–S5. Questionnaires sent to each participant at each time point of stool sample collection (before the trip, after the trip, and at follow-up). Funding proposal to the Gerbert Rüf Foundation. Ethical approval by the ethics committee (EKNZ Project ID 2019-00044).

**Author Contributions:** Study design and experimental concept: E.K., S.B., W.-D.H. and A.E.; writing first draft: M.S., A.M. and A.E.; patient recruitment: E.K. and L.P.; sample processing: M.S., S.H., D.A.T., D.N., O.G., A.G. and J.R.; additional experiments: M.S. and L.L.; statistical and bioinformatic analysis: A.M., T.R., M.M. and H.M.B.S.-S.; manuscript review: M.S., A.M., T.R., E.K., M.M., L.L., H.M.B.S.-S., S.B., W.-D.H. and A.E. All authors have read and agreed to the published version of the manuscript.

**Funding:** This work was supported by the Gebert Rüf Foundation [GRS-060/18] (see Supplementary File S5).

**Institutional Review Board Statement:** The study was conducted in accordance with the Declaration of Helsinki, and approved by the Ethics Committee (EKNZ project ID 2019-00044, on 02/12/2019).

**Informed Consent Statement:** Informed consent was obtained from all subjects involved in the study.

**Data Availability Statement:** Bacterial sequencing data will be shared via publicly accessible databases according to FAIR principles.

**Conflicts of Interest:** The authors do not have a conflict of interest.

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
