# Peer review of "ESBL Displace: A Protocol for an Observational Study to Identify Displacing Escherichia coli Strain Candidates from ESBL-Colonized Travel Returners Using Phenotypic, Genomic Sequencing and Metagenome Analysis"

_2036-7481, doi:10.3390/microbiolres14010015_

Round 1

Reviewer 1 Report

this paper is hard to read, n (we, and our) distributed in all paper. in scientific research do not write , we, our and so on

in material and methods the primer used did not present

40 persons are very low number to make this experiment

in results: only one figure. 1- where DNA and RNA bands

2- Where Genes detected  

Author Response

Dear Reviewer, thank you very much for your comments. We are pleased to provide a response to all comments below. Please let us know if you have any other comments or questions about our work. Kind regards

this paper is hard to read, n (we, and our) distributed in all paper. in scientific research do not write , we, our and so on

  • We had the paper cross checked by two native speakers and we prefer an active writing style instead of a passive one. Is there a specific instruction from the journals side? If MDPI prefers a passive style in its language, please let us know. This would however require a complete rewriting process of the whole manuscript and is not a scientific editing.

in material and methods the primer used did not present

  • We agree with this reviewer that ideally, we would deploy the PCR here. But this article is a study plan and the set of primers has not yet been defined. If there is a specific set the reviewer would recommend, we would be glad about the opinion.

40 persons are very low number to make this experiment

  • To study the prevalence of colonization after travel return and the factors in the questionnaires alone, the number of subjects is indeed small. However, these questions have been addressed in other studies with much larger subject populations. However, our focus is on microbiome analysis and genome sequencing, and for this we nevertheless consider the number of subjects and time points included to be substantial. This is the largest study with this purpose to date.

in results: only one figure. 1- where DNA and RNA bands. 2- Where Genes detected  

  • We are very sorry, but we do not understand the comment of this reviewer? Please provide us more information if you wish to see more.

Reviewer 2 Report

The submitted study protocol, details the methodologies that are being used (and will be used) to conduct a study that might generate a significant amount of valuable data, and that is obviously the greatest strength of the study. The large volume of sequencing data obtained both concerning E. coli genomes and microbiomes over time will be a great resource to the community.

However valuable I find the study, some issues must be addressed by the authors before publication is considered:

Lines 43-46: The authors claim (and I agree) that the spread of infectious agents between humans, animals and environment need to be reduced and for that the chain of transmissions need to be understood. However, it is not clear how their work will achieve this. This should be explained.

Lines 51-52: The authors state that decolonisation only works for individual bacterial species and use it as argument for the need for new approaches. Yet, they are only concerned with one bacterial species - E. coli. The rational behind this association must be clarified. 

Lines 106-108: If the sample collection runs of 52 weeks (i.e., one year) and it started latest in December 2019, the sample collection finished latest on December 2020. Two years passed, why are the authors now submitting the study protocol for publication instead of the results of the study?

Lines 135-136: The reason why the sampling times are not uniformly distributed amongst the 52 weeks should be included.

Line 170: More details about the "specially formulated cryopreservation solution" should be given - either commercial name or composition.

Line 172 (Figure 2): In the text the authors correctly (in my opinion) write that they will select 15 colony forming units, but in the figure they call them strains. Since that up to such point there is no step to test if the colonies are from the same or different strains, the text in the figure should be changed to match the main text. 

Lines 190-191: What are phenotypic evidences or "suspicion" of carbapenemase production? The details should be added to the text.

Lines 195-196: Details of the protocols under evaluation (e.g., what are they detecting) should be given in the text.

Line 213: One assumes from the "UCP PCR water" that the UCP Qiagen kit is being used, but that should be stated.

Line 253-254: More details should be added as this sentence as it is not really informative.

Line 272: Typo?

Line 287: Incomplete sentence?

Lines 289-301: 40 patients were enrolled. 2 dropped out before the initial sample collection. 38 remained. 3 were lost during follow up. 35 remained. Hence, all samples were already collected and processed, correct? Were they processed up to were in the experimental workflow in Figure 2? Day 2? Day 3? Day 4? Were there any progress in any other analysis (e.g., sequencing, PBMCs, displacement)? Why is the progress being so slow (two years passed since sample collection)? With 7 authors involved in sample processing and several of the wet-lab steps automated shouldn't it be further along? What is the reason for this? Were any other constraints encountered that have not been discussed here?

Line 302: The data is shown for 38 patients, but for 3 there are incomplete data because of loss during follow-up. Can the authors explain why these 3 should be included in the presented data?

Line 315: Although used often in the field, MDR should be spelled out on first appearance. 

Line 365-366: Why is this a limitation?

Line 77-78: Nonetheless, no methods for these experiments were included in the study protocol. Are these to be performed in the scope of this study or a wish for follow-up work? This should be clarified in the text. They are part of the proposal added as supplementary material but not included in the main text. Why is that?

- No methods for the PBMC analysis and the displacement experiments were included. And should be, of course.

Author Response

Dear Reviewer, thank you very much for your comments. We are pleased to provide a response to all comments below. Please let us know if you have any other comments or questions about our work. Kind regards

The submitted study protocol, details the methodologies that are being used (and will be used) to conduct a study that might generate a significant amount of valuable data, and that is obviously the greatest strength of the study. The large volume of sequencing data obtained both concerning E. coli genomes and microbiomes over time will be a great resource to the community.

  • We thank the reviewer for this positive feedback.

However valuable I find the study, some issues must be addressed by the authors before publication is considered:

Lines 43-46: The authors claim (and I agree) that the spread of infectious agents between humans, animals and environment need to be reduced and for that the chain of transmissions need to be understood. However, it is not clear how their work will achieve this. This should be explained.

  • In order to address these questions raised by the reviewer, we plan to (I) have a detailed questionnaire be filled out by each study participant. This will allow us to estimate certain risks. However, this is not the main focus of our study. Our idea is to better understand who is colonized with ESBL-PE after the travel and who is not and why. Therefore (II) our driving idea is to reduce the number of colonized patients as potential spreaders of ESBL-PE by accelerating the loss of their colonization status. Therefore, in our study, we evaluate the factors that promote the loss and look for new ways to target decontamination. We have now added an extra sentence to better explain this on line 86-87.

Lines 51-52: The authors state that decolonisation only works for individual bacterial species and use it as argument for the need for new approaches. Yet, they are only concerned with one bacterial species - E. coli. The rational behind this association must be clarified. 

  • We focus on ESBL-producing Enterobacteriaceae. They are among the priority pathogens according to the list of antibiotic-resistant bacteria for which new antibiotics are urgently needed. This list was published by WHO in 2017 (1). The main contributor for the global spread of ESBL-PE is coli as the main vector of this resistance mechanism in a community setting (2-4). New approaches are needed because decolonization with E. coli is usually unsuccessful. We could delete the part of the sentence, "... and work only in single bacterial species." However, we agree with the reviewer, that potentially any colonizing bacteria may be outcompeted with a more fit and adapted strain.

Lines 106-108: If the sample collection runs of 52 weeks (i.e., one year) and it started latest in December 2019, the sample collection finished latest on December 2020. Two years passed, why are the authors now submitting the study protocol for publication instead of the results of the study?

  • The reviewer is clearly right. Participant recruitment lasted until December 2019. Due to the difference in travel dates (participants' last travel was in January 2020, subjects 21 and 22). Therefore, we received our last samples from week 52 in January 2021. Due to the COVID19 pandemic, our human and sequencing resources were temporarily overloaded with the testing and sequencing of SARS-CoV-2 samples. Therefore, our study has been somewhat delayed. However, we heavily work on the first manuscript, but we felt it is very import to publish the study protocol.

Lines 135-136: The reason why the sampling times are not uniformly distributed amongst the 52 weeks should be included.

  • Based on our previous published papers, we knew that the ESBL-PE colonization status shows a strong dynamic within the first weeks after the travel return. Therefore, we hypothesized that the greatest changes in the microbiome and colonization status would tend to occur around the time of travel. We assumed that the microbiome would gradually stabilize after the trip and that relevant changes would then become less likely.

Line 170: More details about the "specially formulated cryopreservation solution" should be given - either commercial name or composition.

  • We use a commercial tube for the single strains, unfortunately, the composition of the cryopreservation solution is not described in detail by the company. For the stool samples we use non-commercial tubes. Our tubes (Eppendorf, 2mL) contain Tryptic Soy Broth (TSB from Sigma-Aldrich, Switzerland) and 15% glycerol. The stool sample is added in a 1:1 ratio. We have added this detail to Line 154-156 of the manuscript.

Line 172 (Figure 2): In the text the authors correctly (in my opinion) write that they will select 15 colony forming units, but in the figure they call them strains. Since that up to such point there is no step to test if the colonies are from the same or different strains, the text in the figure should be changed to match the main text. 

  • The reviewer is right, that this may be confusing. Although, we have started to sequence and also find individual strains and sometimes multiple strains within the same person, this investigation is not finished. Therefore, we agree that we will call them isolates. This is maybe a more neutral term. We have adapted the figure accordingly.

Lines 190-191: What are phenotypic evidences or "suspicion" of carbapenemase production? The details should be added to the text.

  • The phenotypic evidence for ESBL was the specific resistance pattern in the MASTDISC diffusion test. If there was growth on the ESBL screening plate, but the MAST-DISC assay did not showed evidence for an ESBL, we used a PCR to screen for the most common carbapenemase genes. We have further explained this in Line 200-201.

Lines 195-196: Details of the protocols under evaluation (e.g., what are they detecting) should be given in the text.

  • Reviewer 1 asked a similar question. Unfortunately, this is still work in planning and we have not yet established a specific PCR – if Reviewer 2 has own experience with such a PCR we would be happy to integrate this into the paper.

Line 213: One assumes from the "UCP PCR water" that the UCP Qiagen kit is being used, but that should be stated.

  • The reviewer is right, we used the UCP Qiagen kit and we have added this into the manuscript.

Line 253-254: More details should be added as this sentence as it is not really informative.

  • We thank the reviewer for this comment – although the bioinformatic analysis are not finished, we have updated this paragraph which was written during the study design period. We hope that sufficient details are now provided.

Line 272: Typo?

  • We are not sure where a typo is and we have read the sentence again several times. Please let us know in which word you see a typo.

Line 287: Incomplete sentence?

  • We have corrected this sentence.

Lines 289-301: 40 patients were enrolled. 2 dropped out before the initial sample collection. 38 remained. 3 were lost during follow up. 35 remained. Hence, all samples were already collected and processed, correct? Were they processed up to were in the experimental workflow in Figure 2? Day 2? Day 3? Day 4? Were there any progress in any other analysis (e.g., sequencing, PBMCs, displacement)? Why is the progress being so slow (two years passed since sample collection)? With 7 authors involved in sample processing and several of the wet-lab steps automated shouldn't it be further along? What is the reason for this? Were any other constraints encountered that have not been discussed here?

  • We initially began to process the first 2 to 3 time points of all probands, then we moved on to the subjects with the most interesting colonization and loss-of-colonization patterns. From some patients, all time points were already fully processed until storage on day 4, while other patients' frozen stool samples remained unprocessed. We also began sequencing the microbiota and genome of individual strains. Of some strains of which we already had sequencing data and analyses, we also began testing them for their displacement ability in our mouse model. During the pandemic, our team was partially outsourced to sequence the SARS-CoV-2 strains. However, in the past months we made much progress and we now also submit a paper soon.

Line 302: The data is shown for 38 patients, but for 3 there are incomplete data because of loss during follow-up. Can the authors explain why these 3 should be included in the presented data?

  • In order not to lose data on the valuable isolated strains and microbiota of these three subjects, we decided to include them "partially" in our analysis. Of the three subjects for whom all time points could not be collected, two remained negative at all time points collected, one until week 6 (subject 4), the other until week 20 (subject 9). The third (subject 40) was positive for ESBL-PE immediately after the trip. At this point, we had not processed any other time points for this individual. Because we expected the greatest changes in microbiota around the time of travel, we felt it was reasonable not to exclude these subjects.

Line 315: Although used often in the field, MDR should be spelled out on first appearance. 

  • We have corrected the sentence accordingly.

Line 365-366: Why is this a limitation?

  • This is correct – it is not really a limitation, but rather a challenge, as our goal was to observe the reason for decolonization over time, obviously a higher colonization state would be good for our study. We have re-formulated this sentence and thank the reviewer for picking this up.

Line 77-78: Nonetheless, no methods for these experiments were included in the study protocol. Are these to be performed in the scope of this study or a wish for follow-up work? This should be clarified in the text. They are part of the proposal added as supplementary material but not included in the main text. Why is that?

  • We have focused our study protocol on humans. We added a sentence stating that we will publish mouse related work in a different protocol. This would tremendously expand the already long manuscript.

- No methods for the PBMC analysis and the displacement experiments were included. And should be, of course.

  • We agree with this comment. However, the PBMCs were for the moment just collected and no specific protocol has been evaluated. An idea could be to look into the B-cell receptor repertoire and see if specific antibodies are produced against ESBL-PE. But as mentioned these protocols are not yet determined, and we wanted to add the PBMC part rather for completeness. If the reviewer wishes, we can also remove this study material.

Bibliography:

  1. E. Tacconelli  (Infectious  Diseases DC, TübingenUniversity,  Germany)  and  N. Magrini (WHO, EMP Department), Y.  Carmeli,  Tel  Aviv  University,  Israel; S. Harbarth,  University  of  Geneva, Switzerland; G.  Kahlmeter,  University  of  Uppsala,  Sweden;  J.  Kluytmans,  University  Medical  Center Utrecht, Netherlands; M.  Mendelson, Universityof Cape Town, Groote Schuur Hospital, Cape Town, South  Africa; C.  Pulcini,  University  of  Lorraine  and  Nancy  University  Hospital,  France; N.  Singh, George Washington University, USA; U. Theuretzbacher, Center forAnti-infective Agents, Austria. GLOBAL PRIORITY LIST OF ANTIBIOTIC-RESISTANT BACTERIA TO GUIDE RESEARCH, DISCOVERY, AND DEVELOPMENTOF NEW ANTIBIOTICS. WHO: www.who.int; 2017.
  2. Doi Y, Iovleva A, Bonomo RA. The ecology of extended-spectrum beta-lactamases (ESBLs) in the developed world. J Travel Med 2017;24(suppl_1):S44-S51, doi:10.1093/jtm/taw102
  3. Pitout JD, Laupland KB. Extended-spectrum beta-lactamase-producing Enterobacteriaceae: an emerging public-health concern. Lancet Infect Dis 2008;8(3):159-66, doi:10.1016/S1473-3099(08)70041-0
  4. Bezabih YM, Sabiiti W, Alamneh E, et al. The global prevalence and trend of human intestinal carriage of ESBL-producing Escherichia coli in the community. J Antimicrob Chemother 2021;76(1):22-29, doi:10.1093/jac/dkaa399

Round 2

Reviewer 1 Report

Accept in present form

Author Response

We are very pleased to have been able to address all of the questions and comments raised by Reviewer 1.

Reviewer 2 Report

I am satisfied with the majority of the answers provided by the authors but urge them to add the additional clarifications on the text, for the benefit of the reader and not just of the reviewers.

Lines 195-196 (first version): I believe this is just a matter of phrasing. If it is currently under evaluation, the authors are evaluating something and should be able to list what they are currently evaluating. If not currently evaluating anything then the sentence should be something in the lines of "various protocolos will be evaluated.".

Line 272 (first version): Shouldn't it read "alternatively"?

Line 365-366 (first version): It was the third (now second) limitation that needs additional explanations.

Author Response

I am satisfied with the majority of the answers provided by the authors but urge them to add the additional clarifications on the text, for the benefit of the reader and not just of the reviewers.

  • We are very pleased that we were able to answer most of the questions and comments and would like to address the following ones.
  • We have now added the clarifications to our manuscript according to our last answers on the first comments. Changes have been made in:
  • lines 55-59 (latest version)
  • lines 99-103
  • lines 154-159
  • lines 200-201
  • lines 338-342
  • We have also added 4 new references for the inserted passages (references 9-12).

Lines 195-196 (first version): I believe this is just a matter of phrasing. If it is currently under evaluation, the authors are evaluating something and should be able to list what they are currently evaluating. If not currently evaluating anything then the sentence should be something in the lines of "various protocolos will be evaluated.".

  • We thank the reviewer again for this comment. We have corrected this sentence in line 230

Line 272 (first version): Shouldn't it read "alternatively"?

  • We have corrected the typo in line 305.

Line 365-366 (first version): It was the third (now second) limitation that needs additional Explanations

  • We are sorry that we misunderstood the reviewer at first. We have now adjusted this section. We have reinserted the first limitation and adjusted the last one so that it should be more unterstandable now (lines 406-414).